# Could Gwadar Port in Pakistan Be a New Gateway? A Network Simulation Approach in the Context of the Belt and Road Initiative

**Ryuichi Shibasaki [1,\*]** **, Satoshi Tanabe [2], Hironori Kato [3] and Paul Tae-Woo Lee [4]**

[1] Department of Systems Innovation, School of Engineering, The University of Tokyo, Bunkyo, Tokyo 113-8656, Japan

[2] PADECO Co. Ltd., Minato, Tokyo 113-0034, Japan; satoshi.tanabe@padeco.co.jp

[3] Department of Civil Engineering, School of Engineering, The University of Tokyo, Bunkyo, Tokyo 113-8656, Japan; kato@civil.t.u-tokyo.ac.jp

[4] Maritime Logistics and Free Trade Islands Research Center, Ocean College, Zhejiang University, Zhoushan 316021, China; paultaewoo.lee@zju.edu.cn

\* Correspondence: shibasaki@sys.t.u-tokyo.ac.jp; Tel.: +81-3-5841-6546

**Abstract:** Central Asia (CA), comprising six independent countries and the Xinjiang Uygur Autonomous Region of China (XUAR), is an archetypal landlocked region suffering from poor access to global markets. Possible gateway seaports for CA cargo are scattered across the Eurasian continent, and access requires long-haul transport. Thanks to their shorter hinterland transport distances, Pakistani ports, including Gwadar Port, which has drawn attention in the context of China's Belt and Road Initiative, are investing a substantial amount in their infrastructure, with the aim of becoming the new gateway seaport for CA cargo. This paper aims to analyse the market potential of Gwadar Port and other Pakistani ports as gateways of the cargo to/from CA countries, including China and Russia, applying a two-layered network assignment model, developed from the perspective of shippers, under six scenario conditions. To overcome the lack of data availability in the region, surveys and interviews were conducted. The simulation results, based on several policy scenarios concerning the use of Gwadar Port, with hinterland connections and reduced border barriers, show that the port could handle a sustainable number of containers. If the hinterland rail network effectively connected the port to the CA countries via the XUAR, Pakistani ports could become gateways for CA cargo.

**Keywords:** Central Asia; gateway seaports; intermodal transport; container shipping; BRI; Gwadar port; Pakistan; network assignment; hinterland

## 1. Introduction

The Central Asia (CA) region, comprising Afghanistan, Kazakhstan, Kyrgyz, Tajikistan, Turkmenistan and Uzbekistan, is a typical landlocked area that has suffered from limited access to global markets [1,2]. CA is surrounded by Russia, China, Pakistan, Iran and the South Caucasus across the Caspian Sea. Therefore, all of the countries aim to become gateways for maritime cargo to/from the CA countries. While the current major trade partners of the CA countries are those adjacent countries, accessibility to seaports is crucial for CA countries' sustainable economic growth. As Gallup et al. [3] pointed out, the lower economic growth of landlocked countries can be attributed to their limited access to seaports.

Many studies have shown the importance of transport infrastructure for the international trade of CA. China's Belt and Road Initiative (BRI), promoting regional infrastructure integration among Eurasian countries, is also concerned with transport infrastructure that would connect China to Europe. For example, regular container train services (China–Europe Railway Express) that connect Chinese cities (mainly inland cities, such as Chongqing, Chengdu, Wuhan and Xi'an) with European cities, including Russian ones, have been increasing rapidly in both their number and the amount of area that they cover in both China and Europe. The Silk Road Economic Belt concept also involves intermodal routes with land–seaport connectivity, covering Iran, the South Caucasus, Turkey and other European countries via the CA countries, without having to pass through Russia.

Meanwhile, the CA Regional Economic Cooperation (CAREC) programme, led by the Asian Development Bank (ADB), has been promoting regional cooperation and development in CA since 1997. Currently, the CAREC programme has 11 partner countries, comprising not only the CA countries, but also China (focusing on the Xinjiang Uygur Autonomous Region (XUAR)), Mongolia, Pakistan, Azerbaijan and Georgia. From the viewpoint of the international logistics environment, the XUAR is similar to the CA countries, being over 3000 km away from Chinese seaports. Thanks to the best 'physical connectedness' of the port-hinterland [4] in Pakistan and China's Belt and Road Initiative, the country's ports, including Gwadar Port, are investing a substantial amount in their infrastructure, with the aim of becoming the new gateway seaports for CA cargo. In other words, Pakistani seaports (Karachi, Bin Qasim and Gwadar) are becoming promising gateways for CA, including the XUAR, to connect their hinterland, because most cities in CA are closer to the Pakistani seaports than the ports in Russia, China, Iran, Georgia and the Baltic countries. In fact, Pakistan joined the CAREC programme in 2010, with the aim of offering new seaport gateways for the hinterland in CA. However, the current hinterland of the Pakistani seaports extends only to Afghanistan, rather than to Pakistan itself, because of insufficient infrastructure, especially in Afghanistan, which makes it impossible to access the CA. In order for Pakistani seaports to serve the hinterland, many projects to overcome the present constraints in infrastructure in the country are either being planned or have already been implemented. For example, the China–Pakistan economic corridor (CPEC), in the context of BRI, includes launching a new seaport (Gwadar Port), enhancing domestic and international rail connections and reducing the physical and institutional barriers at the national land borders.

As ADB [5] pointed out, most of the access routes to CA used to be for the east–west traffic, rather than the north–south traffic, even though the latter provides a shorter access to the gateway seaports. Pakistan aims to break the status quo in infrastructure investment and cross-border facilitation. While the literature, including the CPEC, has mainly highlighted the traffic analysis and role of the CPEC from the Pakistani perspective, few studies have focused on the gateway role of her seaports for CA, based on a quantitative analysis, to estimate container cargoes at major ports in Pakistan using scenario analysis. This has motivated this study to highlight a potential role of Pakistan in providing seaport gateways for CA because of her geographical advantages, which could support the sustainable development of the CA region. To the best of the authors' knowledge, there have been no successful studies on the choice of transport routes for containers to/from CA to cover any gateway seaports across the Eurasian continent, which is due mainly to the availability of data and is partly because CA is remote in relation to world markets. To overcome the lack of data, the authors conducted a field survey for years, including several national land borders (see Tables A1 and A2 in the Appendix A). This study applies a two-layered network assignment model (NAM), developed from the perspective of shippers [6,7], to analyse the market potential of Gwadar Port and other Pakistani ports, as gateways of the cargo to/from CA countries, including China and Russia, under six scenario conditions. The study covers the entire maritime container cargo for CA, not only China through Pakistani seaports. The authors provide a tool for quantitatively analysing them by incorporating the know-how accumulated by the analyses of different regions of the world.

The remainder of this paper is organised as follows. Section 2 details a literature review. Section 3 presents the current situation regarding the gateways available to CA among the Eurasian ports for maritime containers to/from CA. Section 4 describes the model and its input data and assesses its performance. Section 5 applies an extended two-layered network assignment model, developed from the perspective of shippers, to simulate policies related to infrastructure investment and cross-border facilitation in Pakistan and its neighbouring countries. Finally, Section 6 concludes the paper with further study issues.

## 2. Literature Review

Landlocked countries with a lack of transport infrastructure and poor trade facilitation face disadvantages in terms of connectivity to trade corridors and intermodal transport, causing higher transport and logistics costs [3,8–12]. Specifically, Limao and Venables [8] focused on the overland transport costs of landlocked countries and estimated the elasticity of their trade volume in relation to their quality of infrastructure. Arvis and Raballand [9] highlighted the complexity of the supply chains in landlocked developing countries and clarified their logistics cost structures. Anukoonwattaka and Saggu [10] emphasised the importance of trade policy in landlocked countries, finding that trade barriers, service trade restrictions and a poor trade facilitation performance caused high trade costs in Asian landlocked developing countries. Kashiha et al. [11], analysing the shipment records of several countries in Europe, found that shippers in landlocked countries avoid long-haul transport, readily cross borders and place more value on transport infrastructure. Lim et al. [12] specified key factors affecting transport corridor development in Northern Asian countries, including Mongolia, the only landlocked nation in the region. Landlocked countries recognise the importance of the geo-relationship, in terms of transport and trade corridors, between landlocked economies and their surrounding countries having seaports [13–15]. Some of the aforementioned studies also indicated that landlocked countries often depend on a single transport route, which makes their economies vulnerable, emphasising the importance of transport corridor diversification in enabling them to negotiate sea access. Additionally, some papers focused on port selection and competition for cargo to/from some specific landlocked territories, such as Austria [11,16], Niger [17], Laos [18,19] and the northeast part of the Southern Asian region [20].

CA is a good exemplary and complicated landlocked area in the world. The World Bank [2] categorised CA into regions far from world markets, indicating that it is comparatively difficult to reduce trade costs due to their distant hinterlands. However, CA has been recently considered as an international trade hub and as a trade partner, which is in line with the recent economic growth of its neighbours, such as China, India and Russia [21]. In this context, some studies attempted to highlight their unsustainable trade patterns, represented by their insufficient intra-regional trade and dependence on natural resource-based exports [22,23] and transport corridors [24–27] in CA. Vinokurov et al. [24] focused mainly on the access to CA from the European perspective, discussing the prioritisation of logistics investment policies, as well as the need for an integrated transport system. Kulipanova [25] discussed the reasons for the difficulties in regional cooperation regarding transport, highlighting major physical and nonphysical barriers to international transport. Rodemann and Templar [26] discussed the enablers and inhibitors of the promotion of rail freight transport between Europe and Asia, not only from an economic perspective, but also from political, technical, legal and environmental perspectives. Islam et al. [27] also investigated the same issue through interview surveys with several stakeholders.

Additionally, some studies examined the bottleneck of regional trade supply chains in CA that are adjacent to emerging economies and located it at the centre of the Eurasian continent [9,28]. Several studies have focused on the international logistics of a specific country or route in CA, such as Bulis and Skapars [29], who focused on the Riga port in Latvia, as a gateway for CA cargo, and Regmi and Hanakoka [30], who compared two specific rail routes between South Korea and European Russia. As for the quantitative approach to simulating route choice related to CA, Wang and Yeo [31,32]

evaluated three transport routes for second-hand cars from Korea to CA and the routes from Korea to Kazakhstan via multiple Chinese ports and land borders. Meanwhile, ADB [5] pointed out that, given regional economic development and the accession of Turkmenistan and Pakistan to the CAREC programme, 'an adjustment in emphasis is needed, giving as much importance to north–south and intra-regional links as with the east–west corridors between East Asia and Europe.' In this context, Yousefi [33] highlighted the role of Iran as a gateway for CA cargo.

There are a few studies related to Pakistan. Sayareh [34] compared the performance of Chabahar port in Iran and that of Gwadar port in Pakistan through interviews with stakeholders. Anwar [35] and Masood et al. [36] discussed the potential of Pakistan as a gateway for CA. From the Chinese perspective, Khan [37] focused on the CPEC between the XUAR and Pakistan, including a new rail connection. The CPEC is regarded as an important link and one of the key examples with which to forecast the future development of the BRI [38–40]. Sheu and Kundu [41] and Wang et al. [42] applied the quantitative approach to simulate the choice of routes for importing oil from Middle Eastern countries into China, between the CPEC (via Pakistan), Myanmar and the direct sea route (via the Malacca Strait), although they did not consider container cargo. Shao et al. [43] and Yang et al. [44,45] focused on the rail transport that connects China with countries in the Eurasian continent and their route competitions in the BRI context, but they did not consider Pakistani seaports.

In summary, no studies have quantitatively considered CA's connectivity to seaports in the neighbouring countries in terms of their competitiveness, focusing on Pakistani seaports, including Gwadar Port. This paper aims to fill this research gap.

## 3. Pakistani Seaports as a Gateway of Central Asia

The competitors of Pakistani ports, as gateways for CA cargo, are spread broadly across the Eurasian continent. Figure 1 shows the major gateway seaports and their access routes for international maritime cargo to/from CA. Broadly speaking, they have three directions in six regions, namely, (i) Far East Russia and China to the east, (ii) the Black Sea (including Georgia and Russia) and Baltic Sea (including Russia and the Baltic countries) to the west, which is a component of the Transport Corridor Europe–Caucasus–Asia (TRACECA), and (iii) Iran and Pakistan to the south. While various gateway seaports to/from CA are available across the continent, the transport time and costs associated with the three access routes are considerably high.

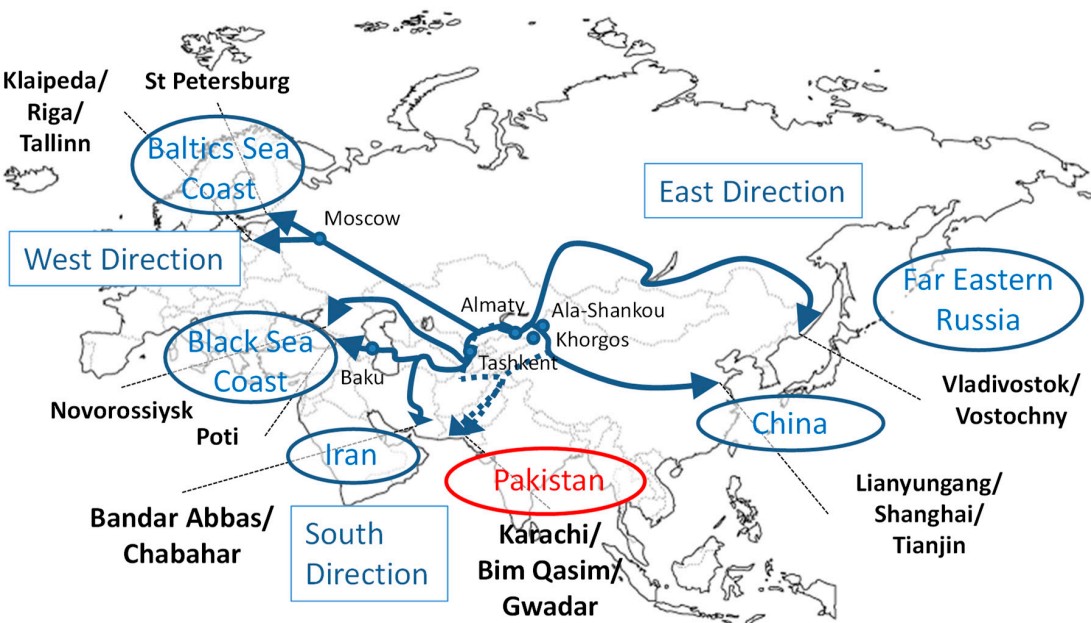

**Figure 1.** Main access routes to the seaports to/from Central Asia (CA). Source: Compiled by the authors.

Among them, the Arabian Sea coast is the nearest coast for almost all CA regions, including the XUAR in China, in terms of the direct distance. However, the route to the Arabian Sea has many national borders and political/geographical difficulties, including unstable and unsafe areas, for instance, in Afghanistan and Baluchistan in Pakistan, and mountainous terrains, for instance, at the border between China and Pakistan. Guaranteeing the security of transport along this access route remains a challenge.

Currently, the sole available gateway seaport in the Arabian Sea, except for the Afghani cargo transported to the Pakistani ports, is Bandar Abbas, which is the largest seaport in Iran, located at the mouth of the Persian Gulf. It takes approximately one week to transport cargo from Tashkent to Bandar Abbas port by truck, which is almost half the time that it takes for the cargo to reach the Chinese coastal ports [46]. It is worth noting that Chabahar port, which is located near the border with Pakistan, is expected to be a future alternative gateway among the Iranian ports.

Pakistani ports, including Karachi, Muhammad Bin Qasim and Gwadar, are becoming attractive for CA countries with respect to their potential function as new gateways, enabling the sustainable development of CA countries, since Pakistan joined the CAREC programme in 2010. However, the hinterland of these ports currently covers only Pakistan and some areas of Afghanistan because of the underdeveloped infrastructure, as well as poor security, when crossing Afghanistan, as already mentioned.

The cargo volume data and specifications of each Pakistani port are summarised in Table 1. The development of hinterland infrastructure is planned, such as a circular railway network (Figure 2) for Afghanistan to connect CA with the infrastructure network of Pakistan and Iran [47].

**Table 1.** Cargo volume handled at the major ports in Pakistan, 2015–2016.

| Port | Cargo Type | Draft | Import | Export | Total | Annual Capacity |
|------|-----------|-------|--------|--------|-------|-----------------|
| | | | Unit (Cargo: Million Tonnes, Container: Million TEUs) | | | |
| Karachi | Cargo | 10–13 m | 40.3 * | 9.78 * | 50.0 * | N/A |
| | Container | 13 m | 1.01 | 0.897 | 1.90 | 1.78 |
| Port Qasim | Cargo | 10–13 m | 25.7 * | 7.44 * | 33.2 * | 63 |
| | Container | 12, 13 m | 0.556 | 0.568 | 1.12 | 2.03 |
| Gwadar | Cargo | 12.5 m | N/A | N/A | 0.00360 ** | N/A |
| | Container | 12.5 m | N/A | N/A | 0.000145 ** | N/A |

* The tonnage of cargo, including that of containers. ** The annual cargo tonnage of the Gwadar Port was reported in 2017. Sources: several studies [48,49], the authors' interview and the website of each port authority.

Currently, approximately 60% of the total international cargo handled by the Pakistani ports is either going to or coming from the Punjab province, far from the Arabian Sea coast (over 1200 km away). However, the railway services in Pakistan are currently insufficient to meet such a transport demand. The current modal share of rail transport accounts for only 4% out of the total freight flow [50]. This very low share of the rail transport mode is caused by (i) the lack of railroad freight cars; (ii) single tracks, even along the main railway line; and (iii) the poor service capacity in terms of port access, not to mention the substandard surface condition of the national roads.

Gwadar Port is a deep-sea port, located in the Balochistan Province of Pakistan, which is 120 km away from the border point with Iran. The initial construction was completed in 2005, and its concessional rights were transferred to China Overseas Ports Holding Company Limited in 2013 for 40 years. The port has been underused, so that it received only 145 TEUs in 2017. The first commercial container shipment was dispatched from Gwadar to UAE in March 2018 [49]. The port development is associated directly with the CPEC [38]. The economic corridor that will connect the port with the XUAR city of Kashi, giving China a direct access to the Arabian Sea, is planned. The corridor

development covers a wide range of transport infrastructure projects, such as linking Gwadar Port with the main artery of the national highway network, developing the motorway between Gwadar city and Karachi city, expanding and reconstructing the existing main railway line in Pakistan and facilitating border-crossing transport infrastructure [51].

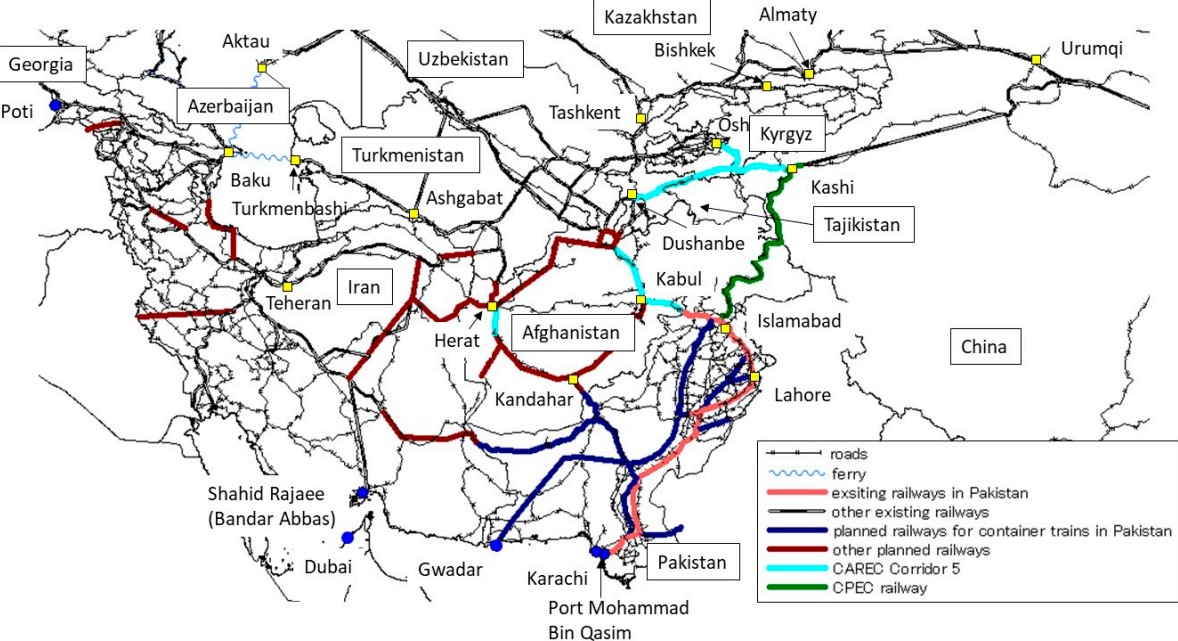

**Figure 2.** Land transport networks connected with the Arabian Sea coastal ports. Source: Compiled by the authors based on the CA Regional Economic Cooperation (CAREC) programme.

## 4. Model

### 4.1. Model Structure

This paper applies a two-layered NAM, developed from the perspective of shippers [6,7] in CA and its surrounding countries, for covering every gateway of the global maritime container shipping (MS) network. Figure 3 shows the entire structure of the model, which consists of a super-network for intermodal transport in the upper layer and two real networks, representing each MS and hinterland transport (HT) network in the lower layer. Only full (i.e., laden) containers are considered, and the regional cargo shipping demand is fixed. Therefore, the impact of some policies or changes in the economic environment in altering the transport demand is beyond the scope of the simulation in this paper; it should rather be considered in other models that can forecast the transport demand (see Shibasaki et al. [52] as an example of applying the international economic model to measure the impact on trade and the cargo transport demand of changes in the economic and transport environment).

The super-network model in the upper layer includes the outputs of the real network submodels in the lower layer, namely, the freight charge and transport time for the MS and HT networks. The MS and HT cargo demands, which are the inputs of the two submodels in the lower layer, are cargo flows of the MS and HT links in the super-network model. There are two major reasons for why the model is divided into two layers. One is that a single-layered network assignment model that incorporates a stochastic approach with a capacity constraint (stochastic user equilibrium (SUE)) is an alternative approach. However, if the Gumbel distribution is assumed for the error terms of the utility function, the results can be affected by the density of the real network, owing to the independent axiom. In contrast, if a normal distribution is assumed, SUE cannot be applied to a huge network. The other reason is that the freight charge is generally decided in the real MS market, based on a combination of export and import ports (i.e., on a path basis in the given MS network), irrespective of the actual

shipping route on the sea. Therefore, different layered networks are necessary to compute both the freight charge and transport cost.

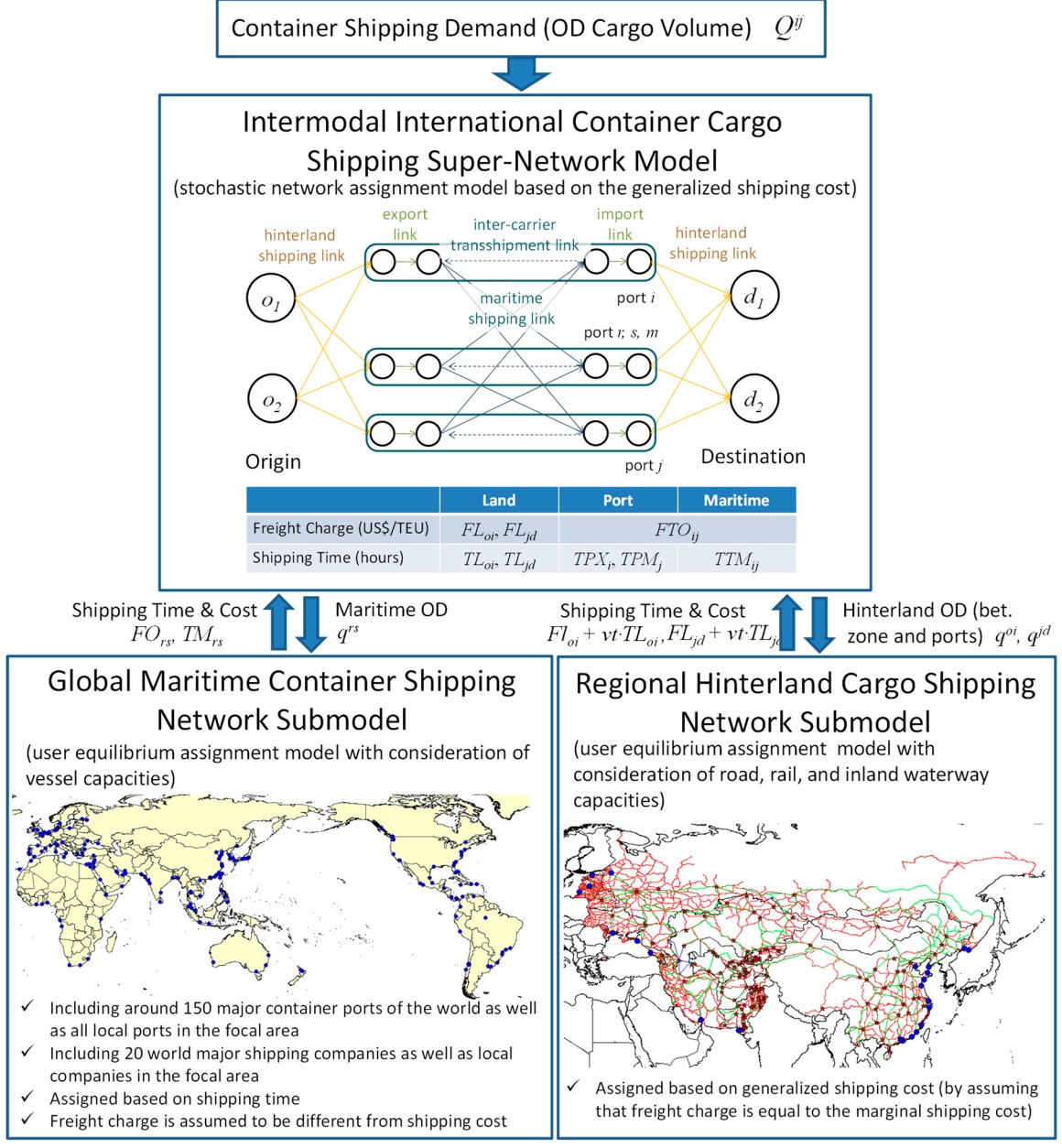

**Figure 3.** The two-layered network assignment model (NAM) structure. Source: Compiled by the authors, based on Shibasaki and Kawasaki [7].

### 4.1.1. Super-Network Model

In the upper layer, each shipper is assumed to choose the ports to be used for export and import, given the freight charges and shipping time for MS and HT on the intermodal network. When $H^{od}$ is the path choice set of the cargo shipping demand $Q^{od}$ (TEU) from origin $o$ to destination $d$, a path $h$ is chosen for a cargo $l$ to maximise utility $U^{od}_{hl}$, including an error term $\varepsilon^{od}_{hl}$. That is,

$$U^{od}_{hl} > U^{od}_{h'l}, \ \forall h,h' \in H^{od}, \ h \neq h', \ (o,d) \in O \times D, \tag{1}$$

$$s.t. \ U^{od}_{hl} = V^{od}_h + \varepsilon^{od}_{hl}, \tag{2}$$

where $O$ is the set of origins; $D$ is the set of destinations; and $V^{od}_h$ is the deterministic term of path $h$, from origin $o$ to destination $d$. If the error term $\varepsilon^{od}_{hl}$ follows a Gumbel distribution, the choice of shipper is formulated as

$$F^{od}_h = Q^{od} \cdot \frac{\exp\left(\theta \cdot V^{od}_h\right)}{\exp\left(\theta \cdot V^{od}_h\right) + \sum\limits_{h' \in H^{od}} \exp\left(\theta \cdot V^{od}_{h'}\right)} \tag{3}$$

where $F^{od}_h$ is the cargo volume on path $h$, from origin $o$ to destination $d$; and $\theta$ is the distribution parameter. The deterministic term $V^{od}_h$ for each path is expressed as the summation of the freight charge and cost related to the transport time:

$$V^{od}_h = -\left(FL_{oi} + FTO_{ij} + FL_{jd}\right) - vt \cdot \left(TL_{oi} + TPX_i + TTM_{ij} + TPM_j + TL_{jd}\right), \; \forall i, j \in h, \tag{4}$$

where $vt$ is the shippers' value of time (US\$/TEU/hour); $FL_{oi}$ and $FL_{jd}$ are the HT freight charge (US\$/TEU), from origin $o$ to port $i$ and from port $j$ to destination $d$, respectively; $TL_{oi}$ and $TL_{jd}$ are the HT time (hour), from origin $o$ to port $i$ and from port $j$ to destination $d$, respectively; $TPX_i$ is the lead time (hour), when exporting from port $i$; $FTO_{ij}$ is the total ocean freight charge (US\$/TEU), from port $i$ to port $j$, including port charges; $TTM_{ij}$ is the total MS time (hour), from port $i$ to port $j$; and $TPM_j$ is the lead time (hours), when importing in port $j$. It should be noted that all port charges are included in the ocean freight charge, $FTO_{ij}$, and any monetary costs are not considered in the port links (i.e., export and import links), since we assume that the ocean freight charge includes not only export and import ports, but also transshipment ports on the way.

Unlike in the study of Shibasaki and Kawasaki [7], inter-carrier transshipment is considered in the model, which is described as a dotted line in Figure 3, for obtaining a more realistic consideration of the actual MS market. It should be noted that intra-carrier transshipment is considered in the global MS submodel. Therefore, unlike in the previous model, the total ocean freight charge, $FTO_{ij}$, and total MS time, $TTM_{ij}$, are described as follows:

$$FTO_{ij} = \sum_{(r,s) \in h} FO_{rs} + \sum_{m \in h, m \neq i, m \neq j} \left(\tau \cdot CR_m - CPX_m - CPM_m\right) \text{ and} \tag{5}$$

$$TTM_{ij} = \sum_{(r,s) \in h} TM_{rs} + \sum_{m \in h, m \neq i, m \neq j} \tau \cdot TR_m \tag{6}$$

where $FO_{rs}$ is the ocean freight charge (US\$/TEU), from port $r$ to port $s$, in path $h$ including port charges; $CR_m$ is the container handling charges (US\$/TEU), when container cargo is transshipped in the same liner shipping company in port $m$; $\tau$ is the multiplier of inter-carrier transshipment ($\tau > 1$); $CPX_m$ and $CPM_m$ are the container handling charges (US\$/TEU), when container cargo is loaded and unloaded, respectively; $TM_{rs}$ is the total MS time (hour), from port $r$ to port $s$ in path $h$; and $TR_m$ is the transshipment time (hour) in the same liner shipping company in port $m$. It should be noted that the transshipment charge, $CR_m$, is usually set between the average and the sum of the loaded and unloaded charges, although detailed information on this for each port is generally not available. Therefore, in this model, we assume that it is as follows:

$$CR_m = 0.75 \cdot \left(CPX_m + CPM_m\right). \tag{7}$$

The ocean freight charge, $FO_{rs}$, and MS time, $TM_{rs}$, are acquired from the calculation results of the MS submodel in the lower layer, while the freight charge, $FL_{oi}$ and $FL_{jd}$, and transport time, $TL_{oi}$ and $TL_{jd}$, of HT are from the HT submodel. Detailed formulations of both submodels are shown as follows.

A cargo flow of each link in this model represents the inputs (i.e., cargo shipping demand) of the submodels in the lower layer, namely

$$q^{rs} = x_{rs},$$ (8)

$$q^{oi} = x_{oi} \text{ and } q^{jd} = x_{jd},$$ (9)

where $q^{rs}$ is the MS cargo demand (TEU/year), from export port $r$ to import port $s$; $x_{rs}$ is the cargo flow (TEU/year) of the MS link; $q^{oi}$ and $q^{jd}$ are the HT cargo demand (TEU/year), from origin $o$ to export port $i$ and import port $j$ to destination $d$, respectively; and $x_{oi}$ and $x_{jd}$ are the cargo flows (TEU/year) of the HT link.

The stochastic assignment model under the intermodal super-network is performed, based on an algorithm proposed by Dial [53].

### 4.1.2. Global MS Submodel

In the lower layer, the MS submodel is defined as a problem of allocating container cargo to the global liner service (GLS) network, based on containership movement data (i.e., the MDS containership databank [54]). Each liner service (LS) network is structured as shown in Figure 4. Each container of the shipper chooses a link from the maritime origin node of an export port to the maritime destination node of an import port. In this submodel, every container of each maritime origin–destination (OD) pair is assumed to choose a route to minimise the total shipping time. The shipper chooses a carrier, considering only the shipping time, not the freight charge. This assumption is based on the idea that the international MS market is oligopolistic, but a freight charge for an OD pair is the same among carriers, if the service is provided and used.

Since vessels for each service have their own capacities, a diseconomy of scale is attained by concentrating on a specific service. Therefore, the congestion of the link is considered, and a user equilibrium (UE) assignment is applied as a network assignment methodology.

$$\min_{x} z(x) = \sum_{a \in A} \int_0^{x_a} t(x_a) dx,$$ (10)

$$s.t. \ x_a = \sum_{(r,s) \in R \times S} \sum_{k \in K^{rs}} \delta_{a,k}^{rs} \cdot f_k^{rs}, \ \forall a,$$ (11)

$$\sum_{k \in K_{rs}} f_k^{rs} - q^{rs} = 0, \ \forall r, s, \text{ and}$$ (12)

$$f_k^{rs} \geq 0, \ \forall k, r, s,$$ (13)

where $a$ is the link; $A$ is the set of links; $x_a$ is the flow of the link $a$; $t_a(.)$ is the cost function of link $a$; $z(.)$ is the objective function; $r$ is the maritime origin; $s$ is the maritime destination; $R$ is the set of the export port; $S$ is the set of the import port; $k$ is the path; $K^{rs}$ is the set of paths for the maritime OD pair $rs$; $\delta_{ak}^{rs}$ is the Kronecker delta; and $f_k^{rs}$ is the flow on path $k$. Kronecker delta, $\delta_{ak}^{rs}$, is written as

$$\delta_{a,k}^{rs} = \begin{cases} 1 \text{ if } a \in k \\ 0 \text{ if } a \notin k \end{cases}.$$ (14)

For a detailed description of the cost function for each link, please see Shibasaki et al. [6].

Regarding networks, only the navigating link has a flow-dependent cost function. The cost functions of the other links are flow independent. According to the UE assignment definition, the MS time, $TM_{rs}$, is defined as

$$TM_{rs} = \min_{k} \left\{ \sum_{a \in k} t(x_a) \right\}.$$ (15)

The ocean freight charge on each MS link, $FO_{rs}$, provided by a carrier is generally different from the monetary cost of the route for the carrier, reflecting the balance of demand and supply on the market. A detailed calculation is also shown in Shibasaki et al. [6]. The UE problem, shown in Equation (10), is solved through the Frank–Wolfe algorithm [55].

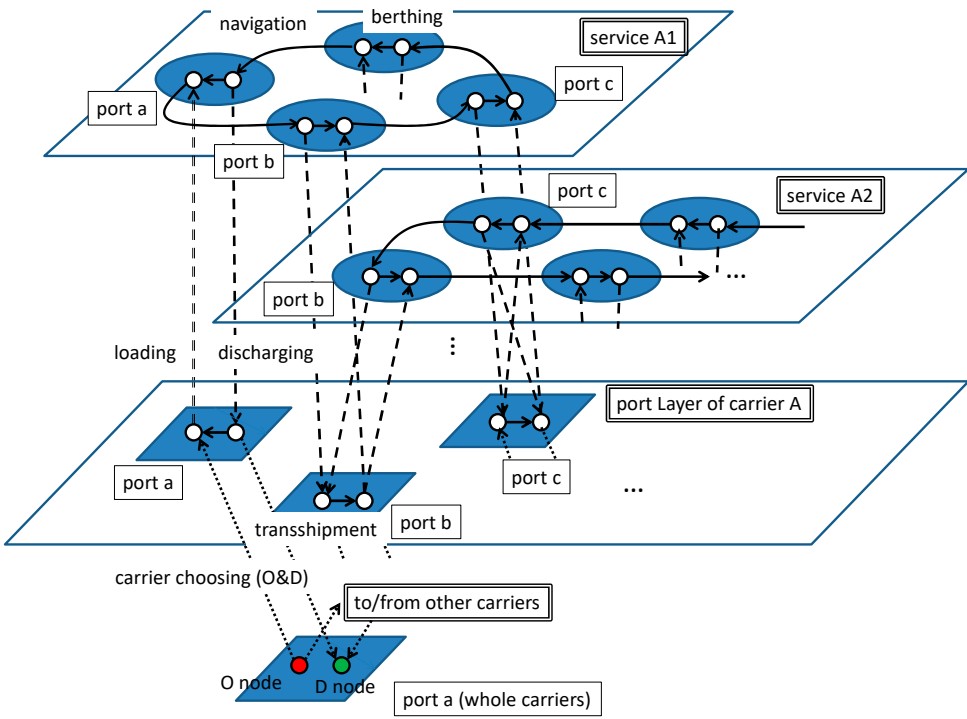

**Figure 4.** Network structure of a maritime container shipping (MS) submodel [6].

### 4.1.3. Regional HT Submodel

Another model in the lower layer is the HT submodel. It is also defined as a problem of allocating container cargo on the HT network in CA and related regions, including roads, rails and inland waterways, with a capacity constraint of each mode. The transport demand, $q^{oi}$ (or $q^{jd}$), between inland origin $o$ (or destination $d$) in the CA region and export port $i$ (or import port $j$) is given. The model is defined as the UE problem, like the MS submodel. It should be noted that, since many enterprises, such as truck companies, are considered to participate in the HT market, the market is sufficiently competitive. Therefore, the shipper chooses the transport mode and route to minimise the total generalised cost, including not only the transport time, but also the freight charge. Namely,

$$\min_{x} z'(x) = \sum_{a \in A} \int_{0}^{x_a} u(x_a) dx, \tag{16}$$

$$s.t. \ x_a = \sum_{(o,i) \in O \times I} \sum_{k \in K^{oi}} \delta_{a,k}^{oi} \cdot f_k^{oi} + \sum_{(j,d) \in J \times D} \sum_{k \in K^{jd}} \delta_{a,k}^{jd} \cdot f_k^{jd}, \ \forall a, \tag{17}$$

$$\left( \sum_{k \in K^{oi}} f_k^{oi} + \sum_{k \in K^{jd}} f_k^{jd} \right) - \left( q^{ir} + q^{sj} \right) = 0, \ \forall o, d, i, j, \text{ and} \tag{18}$$

$$f_k^{oi} \geq 0, f_k^{jd} \geq 0, \ \forall o, d, s, i, j, \tag{19}$$

where $u(.)$ is the cost function of each link; $z'(.)$ is the objective function; $I$ is the set of the export ports; and $J$ is the set of the import ports. It should be noted that the cost function of each HT link, $u_a(.)$, is defined as a generalised cost, not the transport time.

The network structure of the HT submodel is shown in Figure 5. Road and rail networks are connected with a rail connection link, while a ferry link is directly connected with a road or rail link. It should be noted that the cargo origin and destination are connected with only road networks by an OD link, but not by rail, since the last one mile of container transport should be served by a trailer. Another point is that additional transport and time costs are added to each cost function if a road, rail or ferry link crosses the national border, considering the border barrier effect.

Regarding networks, each road, rail and ferry link has different flow-dependent cost functions, as shown in Appendix A. According to the UE assignment definition, the generalised cost of HT, $GL_{oi}$ (or $GL_{jd}$), is defined as

$$GL_{oi}\left(or\ GL_{jd}\right) = \min_k\left\{\sum_{a\in k} u(x_a)\right\} \tag{20}$$

These generalised costs are related to the freight charge and transport time of HT, which are included in Equation (4) and can be expressed as

$$GL_{oi} = FL_{oi} + vt \cdot TL_{oi} \text{and} \tag{21}$$

$$GL_{jd} = FL_{jd} + vt \cdot TL_{jd} \tag{22}$$

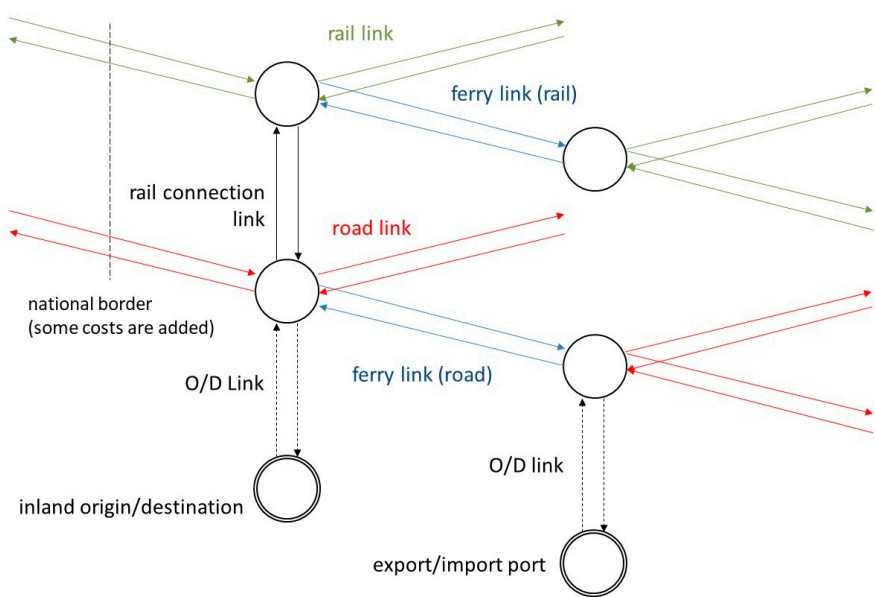

**Figure 5.** Network structure of a hinterland transport (HT) submodel [7].

*4.2. Data*

4.2.1. Ports

The GLS network formed by the major liner shipping companies is covered by the NAM model. In principle, all container ports whose international throughput was over 500,000 TEU per year, as of 2013 (including empty containers but excluding domestic containers), are included [6]. Additionally, the model includes five gateway seaports (i.e., Vostochny in Russia, Poti in Georgia, Klaipeda in Lithuania, Riga in Latvia and Tallinn in Estonia), which were mentioned in Section 3, although their throughput is less than 500,000 TEU, as of 2013. It should be noted that Gwadar Port is not included in the current GLS network, because container throughput data were unavailable for 2013. Furthermore,

several local container ports along the Arabian Sea and the Indian Ocean, such as Sohar in Oman, are included, because we later focus on the CA gateway seaports along the Arabian Sea in the scenario analyses. Consequently, the total number of container ports considered in the NAM model is 187.

The lead times at a terminal for exports and imports, and the handling charges at a container terminal for exports and imports, were set by country, following the 'Doing Business—Trading Across Borders' website [56]. The transshipment time for each port is estimated by evaluating the comprehensive level of service in each port [6].

### 4.2.2. Global MS Network

The MS network is developed based on the MDS containership databank data [54]. Because the model focuses on the container flow in the GLS network and the transshipment of containers at hub ports, some liner services provided by smaller local companies are eliminated from the network for the sake of computational simplicity [6]. Specifically, the model includes the 20 largest liner shipping companies in the world, as of 2013, as well as 17 local companies that have a liner service network at CA gateway seaports. Consequently, 1018 services are included in the model, covering 72.1% of the global annual vessel capacity and 86.7% of the annual capacity of vessels that call at the gateway seaports of CA.

### 4.2.3. Eurasian HT Network

The HT network covers 19 countries in the Eurasian continent, including all CA countries and all potential gateway seaports for CA cargo, as shown in Figure 6. After all road networks and selected rail links are extracted from the ADC WorldMap [57], international ferry links in the Caspian Sea, connecting Baku (Azerbaijan) with Turkmenbashi (Turkmenistan) or Aktau (Kazakhstan), are added, including both road and rail ferries.

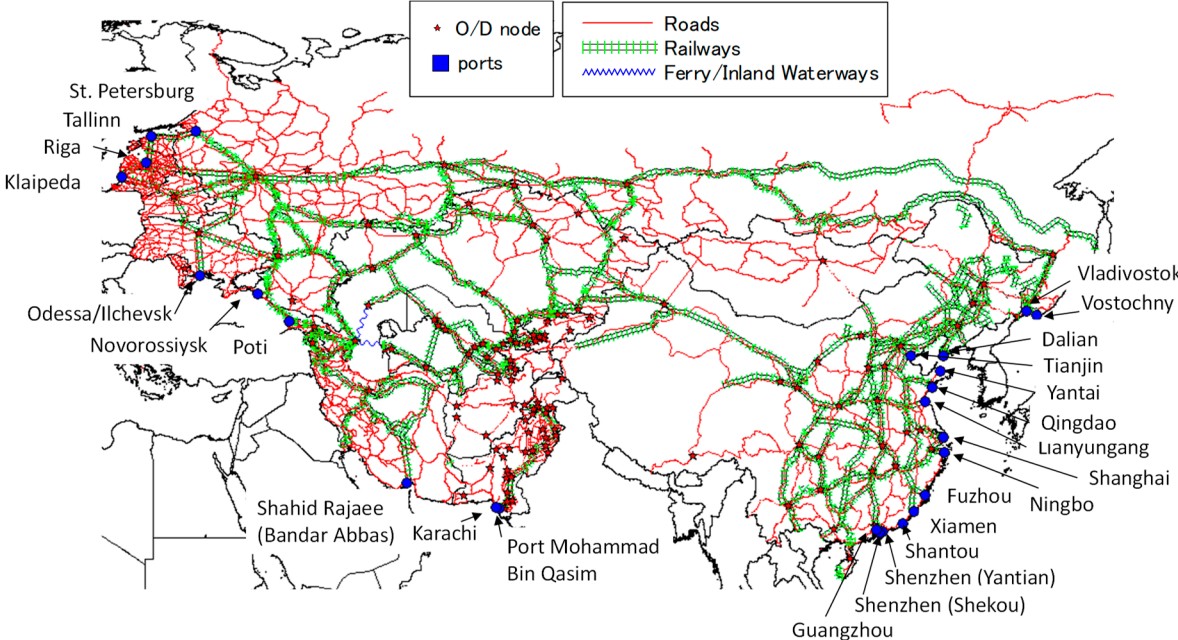

**Figure 6.** HT network in the NAM. Source: Compiled by the authors, based on ADC WorldMap [57].

### 4.2.4. Container Shipping Demand between Regions (Container OD Cargo)

The shipping demand for container cargo (container OD cargo), as well as the initial MS demand, is estimated using various existing data sources, which is the same methodology used in the previous models [6,7].

Firstly, the demand for container cargo shipping (OD matrix) between countries or regions on a TEU basis is obtained from the World Trade Service (WTS) data [58]. The WTS data provide the container shipping demand among 116 countries/regions of the world in 2013. Next, after aggregating the OD matrix into 64 countries/regions to integrate certain countries, they are divided again into a port-based OD matrix, with each port's share of the local container cargo throughput in the aggregated regions.

Then, a region-based OD matrix is estimated for nineteen countries in the Eurasian continent. The port-based OD matrix to/from these countries is again aggregated into a country-based OD matrix (Note that both the breakdown of CA into Kazakhstan, Kyrgyz, Tajikistan, Turkmenistan and Uzbekistan and that of South Caucasus into Armenia, Azerbaijan and Georgia are conducted using the data specially provided by IHS, Inc., from 2011, according to the share by partner country/region. Additionally, the shipping demand to/from Afghanistan is separated from those to/from Bhutan and Nepal using the exported and imported shares in terms of the trade amount (on a value basis), as acquired from the United Nations Comtrade database [59]). Subsequently, we divide it into sub-country levels (zones), such as provinces, federal districts and oblasts, based on the available statistics for the economy of each zone (see Table 2).

**Table 2.** Zonal division into the sub-country level and representative economy indices by the zone of each country in the HT submodel.

| Country | Number of Sub-countries (zones) | Zone Level | Representative of Zonal Economy | Source |
|---|---|---|---|---|
| Kazakhstan | 14 | Oblast level | Import: zonal value of imports (2013) Export: zonal value of manufacturing production (2013) | [60] |
| Kyrgyz | 8 | Oblast level | Gross Regional Product (2012) | [61] |
| Tajikistan | 5 | Province level | Gross Regional Product (2012) | [62] |
| Uzbekistan | 13 | Province level | Gross Regional Product (2012) | [63] |
| Turkmenistan | 6 | Province level | Population (2001) | [64] |
| Afghanistan | 7 | United Nation region level | Population (2014–2015 estimates) | [65] |
| Pakistan | 31 | Division level | Gross Regional Product (2000) at the province level and population (1998) at the division level | [66,67] |
| Russia | 8 | Federal district level | Gross Regional Product (2009) | [68] |
| China | 31 | Province level | regional value of exports and imports (2014) | [69] |
| Armenia, Azerbaijan, Belarus, Estonia, Georgia, Iran, Latvia, Lithuania, Moldova, Mongolia and Ukraine | 1 | Country level | - | - |

Another point to note is that the shipping demand among European countries (including Russia, CA (except Afghanistan) and the South Caucasus) are not considered in the model, because they are not included in the WTS data. In other words, any cargo between European countries (e.g., between CA and Western Europe) is assumed to be transported by land, rather than by MS.

### 4.3. Model Performance

#### 4.3.1. Port Throughput

The developed two-layered NAM is validated from the perspective of its agreement with the observed data. These performance checks are important, especially for a huge simulation model that has many parameters. We also confirmed that the model computation converges well (see Shibasaki and Kawasaki [7,70] for detailed information). For example, Figure 7 indicates the agreement of the model-estimated annual rate and number of laden transshipment containers in major world hubs, with the observed number. These figures show the high multiple correlation coefficients in terms of both the transshipment rate and throughput.

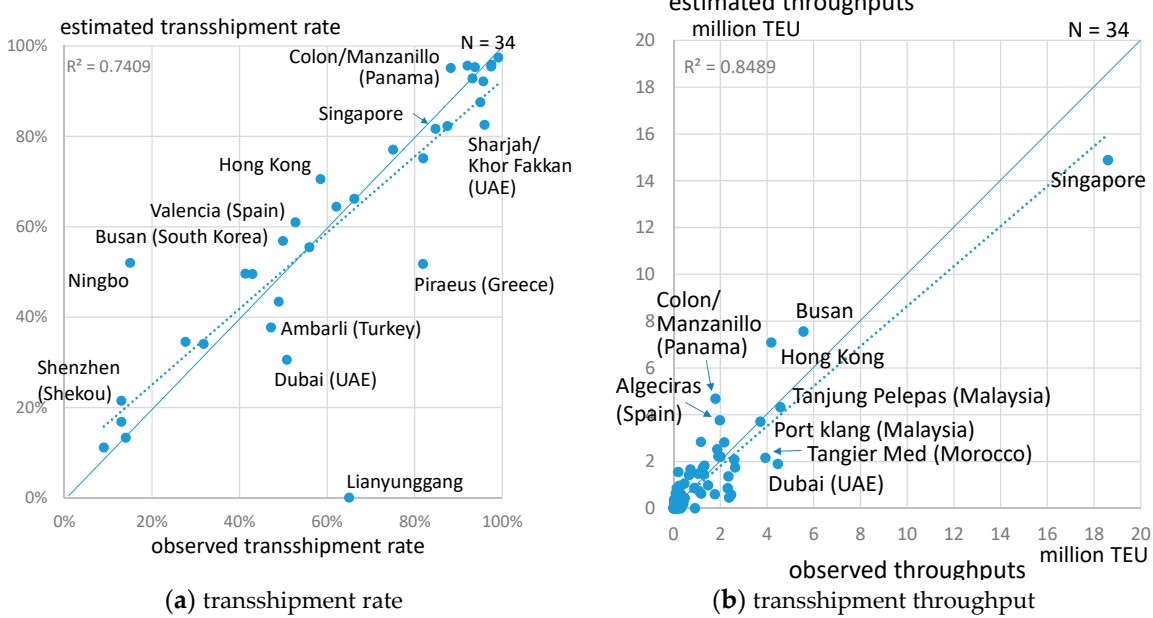

**Figure 7.** Comparison of the observed and model-estimated transshipped containers at global hub ports. Source: Compiled by the authors.

Figures 8 and 9 compare the observed export and import container throughputs with the model-estimated ones for all the CA gateway seaports. The charts on the left of both figures show the throughputs in all 25 seaports, which are included in the regional HT submodel. It should be noted that some Chinese ports (i.e., (i) Qingdao and Yantai; (ii) Shanghai and Ningbo; (iii) Fuzhou, Xiamen and Shantou; and (iv) Shenzhen (two terminals) and Guangzhou ports) are integrated in the figures; the ports categorised in the same group are located so closely to each other that breaking the throughputs down by each port is beyond the scope of the model. The charts on the right of both figures show the throughputs of twelve seaports, none of which are Chinese ports, because the throughputs of Chinese ports are so large, relative to those of other ports, that comparing the latter with the Chinese ports would be impractical.

From these figures, the model accurately estimates the observed throughputs of export and import containers for each gateway seaport. The observed throughputs are different from the model-estimated ones for several Chinese ports, such as Shenzhen–Guangzhou and Qingdao–Yantai, for exports, and

Tianjin and Shanghai–Ningbo for imports. These should be improved by focusing on Chinese cargo and dividing the country into more detailed zones. Additionally, especially regarding exports, some trade-offs between the observed and model-estimated throughputs are observed among the closely located ports, such as Karachi versus Bin Qasim in Pakistan and Vladivostok versus Vostochny in Far Eastern Russia. It may be difficult to differentiate these ports precisely under the current zoning system, because both are located in the same zone. A more detailed zoning system is therefore required for making such a differentiation more reasonable.

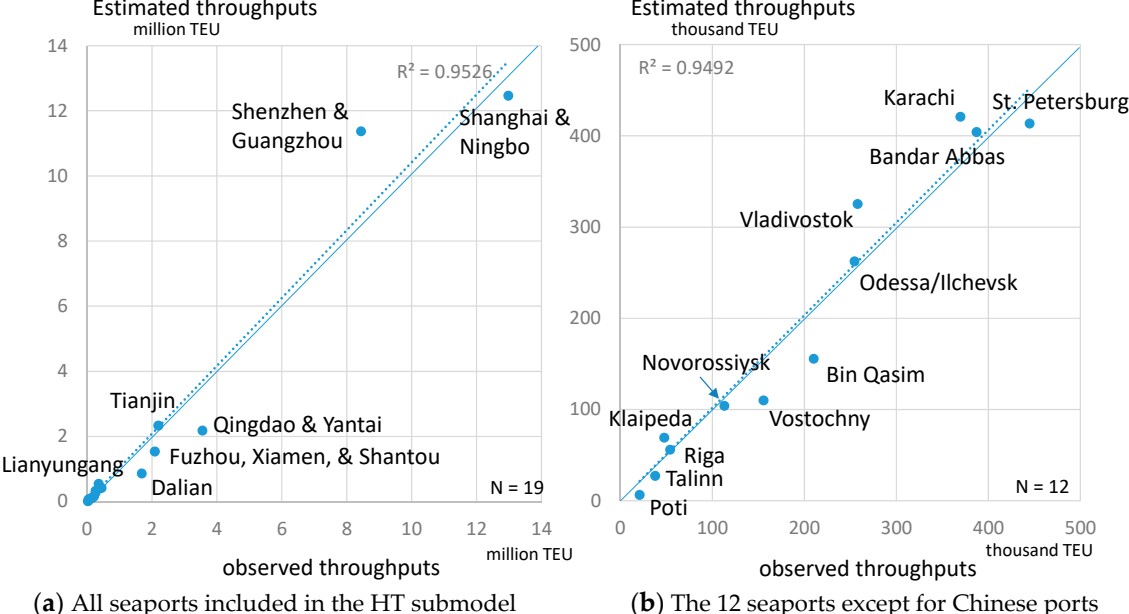

(**a**) All seaports included in the HT submodel    (**b**) The 12 seaports except for Chinese ports

**Figure 8.** Comparison of the observed and model-estimated export container throughputs of gateway seaports. Source: Compiled by the authors.

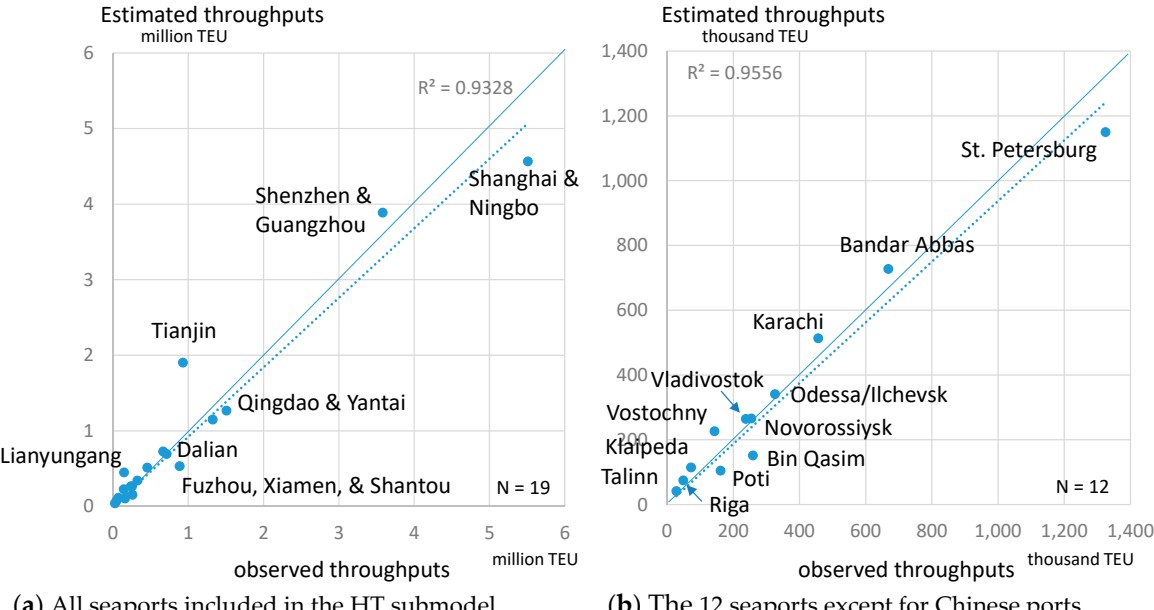

(**a**) All seaports included in the HT submodel    (**b**) The 12 seaports except for Chinese ports

**Figure 9.** Comparison of the observed and model-estimated import container throughputs of gateway seaports. Source: Compiled by the authors.

4.3.2. Choice of Gateway Seaport

Figure 10 shows the breakdown of gateway seaports by partner regions for the selected countries/regions. It should be noted again that the model does not include the container shipping demand of CA (including Russia but excluding Afghanistan) and Europe.

The finding is that the gateway seaports significantly differ according to both the CA country and its partner regions. For example, Figure 10 shows that the containers transported between the Siberian Federal District (SFD) of Russia and (North- and South-) East Asia/South Asia/the Middle East (referred to hereinafter as the eastern regions) are mainly transported via the Far Eastern Russian ports, whereas those transported between the SFD and the Near East/Africa/the American continent (the western regions) are mainly transported via the Black Sea ports and Baltic Sea ports. The same finding can be observed in the containers transported to/from Kazakhstan and Kyrgyz (see Figure 9), although those transported to/from the eastern regions mainly use Chinese ports, instead of the Far Eastern Russian ports. Additionally, some containers transported between these countries and African/American continents are transported via Chinese ports. In summary, the choice of gateway seaports for the SFD, Kazakhstan and Kyrgyz cargo mainly crosses the Eurasian continent in the east–west direction.

The choice of gateway seaports for the containers transported to/from the other four CA countries (i.e., Tajikistan, Uzbekistan, Turkmenistan and Afghanistan) seems to be different from that in the above countries and regions. The main gateway seaports for such cargo are the Pakistani ports (Karachi and Muhammad Bin Qasim), one Iranian port (Bandar Abbas) and the Black Sea ports (Novorossiysk and Poti). Specifically, most Uzbekistani containers are transported via the Black Sea ports, even though they come mostly from Northeast Asia, except for those transported to/from South Asia using an Iranian port, as shown in Figure 10. A similar trend is observed in the containers transported to/from Tajikistan, while some containers transported to/from the eastern regions use Pakistani ports, which is different from the Uzbekistani containers. The usage pattern of gateway seaports for containers transported to/from Turkmenistan is different from that of the above countries; the latter mainly use an Iranian port, and only containers transported to/from the Near East (Turkey is one of the most important partners for Turkmenistan) use Black Sea ports. Furthermore, the containers exported from/imported to Afghanistan mainly use Pakistani ports, but sometimes an Iranian port. This choice is based on the region within Afghanistan, rather than the partner regions: most containers to/from the western regions of Afghanistan (i.e., Herat and Farah) use an Iranian port, as shown in Figure 11.

In conclusion, most of these above-mentioned differences in gateway seaports among the CA countries (including the XUAR and SFD) and trade partner regions seem reasonable, as well as consistent with our qualitative observations, introduced in Section 3, although the estimation results cannot be quantitatively compared with the observed data because of the imperfect data availability. A significant exception is that the share of Chinese ports in the containers imported to Uzbekistan seems very small, although containers imported from the Far East constitute a significant portion of it. According to our interview survey in Uzbekistan, the share of Chinese ports should be larger than that estimated by the model. This difference implies that Uzbek cargo is the most difficult to model, because it is one of two double-landlocked countries (a landlocked country surrounded only by landlocked countries) in the world.

The other main difference in our estimation results is that some Tajik containers use Pakistani ports, whereas the actual hinterland of Pakistani ports is currently only part of Afghanistan, as acquired from the interview of Pakistani forwarders. This may be because the model insufficiently considers the unsafe conditions for logistics within Afghanistan. In other words, the model estimation results suggested a potential of Pakistani ports as gateways for not only Afghanistan, but also other CA countries, while the containers transported to/from the XUAR in China do not use Pakistani ports under the present conditions.

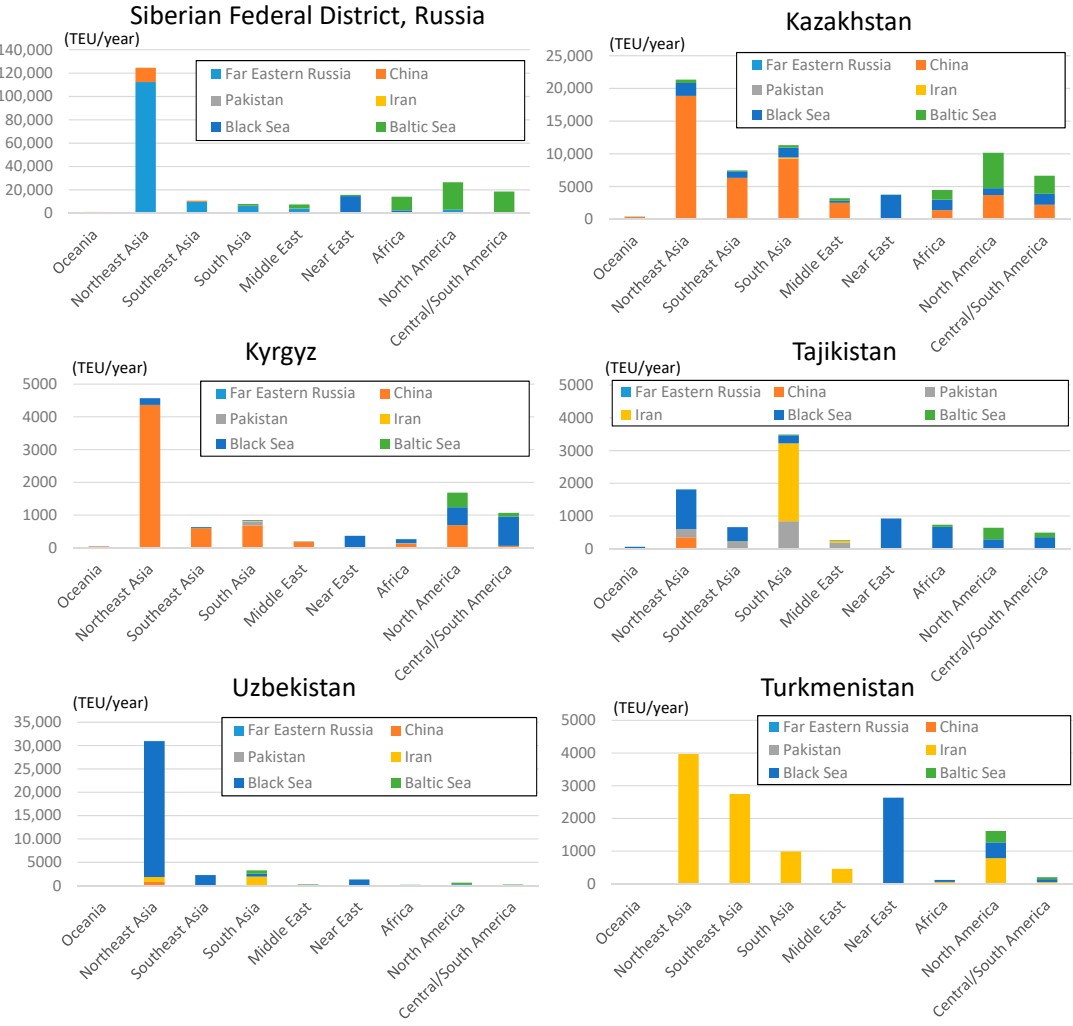

**Figure 10.** Breakdown of the container throughput at gateway seaports by partner region and by country. Source: Compiled by the authors.

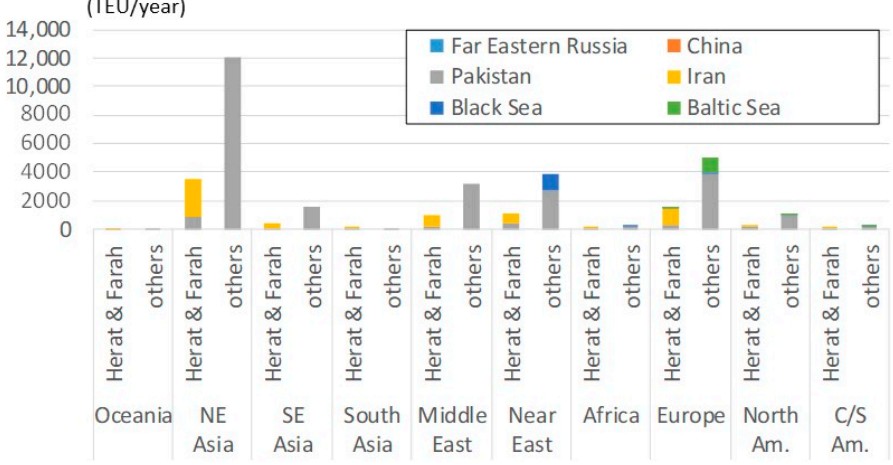

**Figure 11.** Breakdown of the container throughput at gateway seaports by partner regions for Afghan containers. Source: Compiled by the authors.

## 5. Scenario Analysis for the Pakistani Intermodal Network

### 5.1. Scenarios Prepared

Since its accession to the CAREC programme in 2010, Pakistan has implemented several policies related to its ports becoming gateways for the sea trade of the CA countries (including the XUAR in China), such as developing Gwadar Port and investing in infrastructure along the CAREC corridors and the CPEC, as partly described in Section 3. The scenarios to be examined in this section involve the operation of Gwadar Port and improvement of the rail connections within Pakistan and with its neighbour countries, as described in Table 3. A base scenario (*s0*) considers the original parameter settings prepared for this model—the results of which are validated in Section 4.3. In all of the following scenarios, including scenario *s1*, we assume that all regional (i.e., within the Arabian Sea, including off the coast of East Africa) LSs on the way call at Gwadar Port. Specifically, among over 1000 LSs globally, twelve services that call at any of the other Pakistani ports (i.e., Karachi and Bin Qasim), as well as any of the Persian Gulf ports (e.g., Bandar Abbas, Dubai and Khor Fakkan), are assumed to call at Gwadar Port. Additionally, the port is assumed to be connected with its hinterland by road in all of the scenarios.

**Table 3.** Scenarios involving the intermodal network improvement in and around Pakistan.

| Scenario Code | Scenario Condition | Description |
|---|---|---|
| *s0* | Base scenario | In addition to the original settings prepared in this model, the changes in the border barriers since 2013 are considered |
| *s1* | Gwadar Port opening | In addition to *s0*, the opening of Gwadar Port (12 services calling), with hinterland connection by road, is considered |
| *s2* | Rail construction within Pakistan | In addition to *s1*, the increase in the service frequency of the existing rail network (from 5 to 50 trains/week) and new rail construction (10 trains/week) in the whole of Pakistan, except for the CPEC, are considered |
| *s3* | International rail connection with neighbouring countries | In addition to *s2*, the opening of the planned railway along the CAREC corridor 5, outside Pakistan, except for the CPEC (10 trains/week), and other planned railways in Afghanistan, Iran and the South Caucasus (at the same level of frequency as the existing rail service in each country) are considered |
| *s4* | The CPEC railway opening | In addition to *s3*, the opening of the CPEC railway (10 trains/week) is considered |
| *s5* | CAREC border reduction | In addition to *s4*, the CAREC border barriers decline to half of the current level is considered |
| *s6* | CAREC border removal | In addition to *s4*, the CAREC border barriers decline to zero is considered |

Source: Compiled by the authors.

Additionally, scenario *s2* to *s6* assume the development of the rail network in and around Pakistan. Scenario *s2* assumes an improvement of the capacity in the existing rail network by increasing the service frequency, as well as a new rail construction throughout Pakistan, except for the CPEC, based on the development plan proposed by the Pakistani government (see Figure 2). Scenario *s3* additionally

assumes the launch of services in the planned railway along the CAREC corridor 5, outside Pakistan (except for the CPEC), as well as those in other planned railways in Afghanistan, Iran and the South Caucasus, as shown in Figure 2. Scenario *s4* assumes the opening of the CPEC railway, in addition to scenario *s3*. Two additional scenarios assume that the CAREC border barriers decline to half of the current level (*s5*) and to zero (*s6*), in addition to scenario *s4*.

## 5.2. Port Throughputs and HT Flows Estimated

Table 4 summarises the estimated annual throughputs of export and import containers for Gwadar Port in each scenario and provides breakdowns by origin/destination country in the hinterland. Table 5 shows the changes by region in the annual throughputs of export and import containers, estimated in the six scenarios, compared to those estimated in the base scenario (*s0*). The results of scenario *s6* are regarded as a reference, because the model computation does not converge with the given number of maximum iterations, namely, ten.

Table 4 indicates that the container throughputs of Gwadar Port are estimated to exceed 10,000 TEUs for both exports and imports, if the railways are connected with the port (*s2* onward). The results of scenario *s2* and *s3* also indicate that most containers using Gwadar Port are either going to or coming from Pakistan, while some import containers are going to Afghanistan and other CA countries (mainly to Tajikistan). Table 5 shows that some containers using Gwadar Port are shifted from other Pakistani ports, because the regional throughput for each region does not change significantly, compared to scenario *s0*.

In contrast, in scenario *s4*, assuming the launch of the CPEC railways, we find not only that the container throughputs of Gwadar Port increase by over 20,000 TEUs for both exports and imports, but also that the total throughputs of Pakistani ports increase by approximately 20,000 TEUs for both exports and imports, shifting from Chinese ports and the Black Sea ports. Particularly, the import containers that shifted to Pakistani ports in scenario *s4* are mainly going to China (i.e., the XUAR) and Northern CA countries (i.e., Kazakhstan and Kyrgyz). Figure 12 shows the differences in container flow estimated in scenarios *s4* and *s0*, suggesting not only shifts in containers from road to rail in and around Pakistan, but also shifts from the east–west direction (to China or European Russia) to the north–south direction between Siberian Russia and Pakistani ports via Kazakhstan and China.

**Table 4.** Estimated annual throughputs of export/import containers in Gwadar Port by origin/destination country in the hinterland by scenario (TEU).

| | Export | | | | | | Import | | | | | |
|---|---|---|---|---|---|---|---|---|---|---|---|---|
| | *s1* | *s2* | *s3* | *s4* | *s5* | *s6* * | *s1* | *s2* | *s3* | *s4* | *s5* | *s6* * |
| Russia | 0 | 0 | 0 | 254 | 928 | 5431 | 0 | 0 | 0 | 1157 | 1246 | 16,697 |
| China | 0 | 0 | 0 | 917 | 4210 | 6479 | 0 | 0 | 0 | 2094 | 2749 | 2963 |
| CA countries (except Afghanistan) | 0 | 39 | 63 | 207 | 369 | 565 | 0 | 1689 | 2937 | 8030 | 8929 | 10,917 |
| Afghanistan | 0 | 389 | 483 | 798 | 631 | 484 | 0 | 2568 | 3015 | 3082 | 3106 | 3769 |
| South Caucasus Countries | 0 | 0 | 0 | 0 | 0 | 1068 | 0 | 0 | 0 | 0 | 0 | 2554 |
| Pakistan | 2130 | 17,562 | 17,115 | 21,500 | 20,695 | 22,074 | 2410 | 8099 | 8171 | 9595 | 9886 | 18,687 |
| Total | 2130 | 17,990 | 17,661 | 23,676 | 26,832 | 36,101 | 2410 | 12,356 | 14,123 | 23,957 | 25,916 | 55,587 |

* not converged. Source: Compiled by the authors.

Additionally, the total container throughputs of Pakistan and those of Gwadar Port, in particular, increase significantly with the decline of border barriers, as shown in scenarios *s5* and *s6*. In these scenarios, even some containers to/from Russia and the South Caucasus use Pakistani ports, in addition to the majority of containers to/from Tajikistan, Uzbekistan, Turkmenistan and Afghanistan.

**Table 5.** Estimated changes in annual throughputs by region and by scenario, compared to the base scenario (*s0*).

| | Export | | | | | | Import | | | | | |
|---|---|---|---|---|---|---|---|---|---|---|---|---|
| | *s1* | *s2* | *s3* | *s4* | *s5* | *s6* * | *s1* | *s2* | *s3* | *s4* | *s5* | *s6* * |
| Far Eastern Russian Ports | −201 | −22 | −233 | −2141 | −19,646 | −23,954 | −357 | −134 | −275 | −4320 | −8963 | −94,906 |
| Chinese Ports | 341 | 120 | 7 | −10,395 | −4309 | −5021 | 171 | −1089 | 41 | −16,387 | −23,438 | 105,850 |
| Pakistani Ports | −207 | 1,126 | 5728 | 18,317 | 37,513 | 120,739 | 402 | 6361 | 16,694 | 43,680 | 86,489 | 292,128 |
| Iranian Port | −1651 | −2127 | −2935 | −2977 | −3398 | −14,699 | 2103 | −3805 | −1709 | −2508 | −10,206 | −67,781 |
| Black Sea Ports | 2218 | 103 | −709 | −2654 | −4183 | −24,910 | 3012 | −6035 | −14,424 | −19,624 | −41,613 | −28,549 |
| Baltic Sea Ports | −500 | 800 | −1858 | −151 | −5977 | −52,155 | −5331 | 4702 | −328 | −840 | −2267 | −206,742 |

* Not converged. Source: Compiled by the authors.

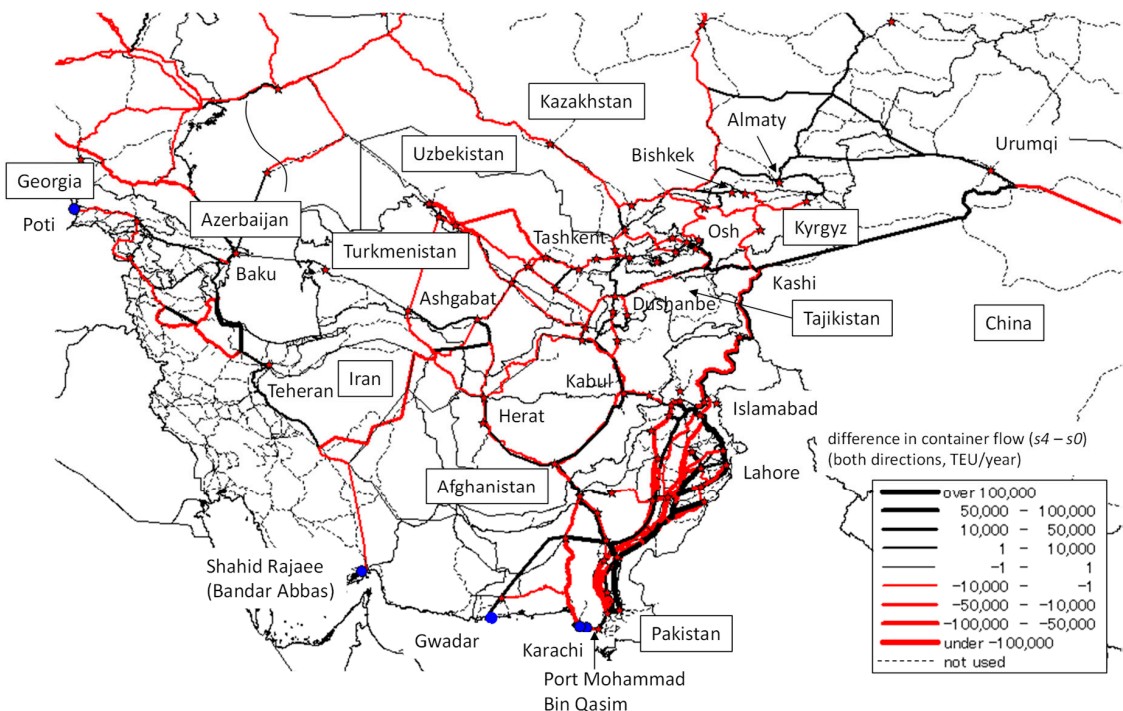

**Figure 12.** Estimated container flow differences between scenario s4 and the base scenario (*s0*). Source: Compiled by the authors.

## 5.3. Estimated MS Flows

Gwadar Port could handle over 30,000 TEUs per year (or 500 TEUs per week) in total, for both exported and imported laden containers, if the rail connectivity between Gwadar Port and each area of Pakistan was well developed and its capacity was sufficient (in scenario *s2* onward). Since it is often

said that 100 to 200 TEUs per week is necessary to maintain one LS to call, the above scenarios would enable several LSs (but not all LSs considered in the simulation) to sustainably call at the Gwadar Port.

Figure 13 shows the regional breakdown of the partner ports for maritime containers handled in three Pakistani ports in scenarios *s2* and *s6*. It should be noted that this is estimated based on the total throughputs, including not only exports and imports, but also transshipment containers, although they share only small portions of the total amount. It should also be noted that the partner port is defined as a port, where containers are finally discharged or first loaded into the LS network of a single company, because of the network structure of the super-network model (see Figure 3). Specifically, the partner port is an export or import port in most cases, but some containers are occasionally transshipped there into a different company's LS.

Figure 13 reveals that the regional shares of partner ports vary across Pakistani ports, while they are not significantly different among scenarios, including other scenarios that are not shown in the figure. The main partners of Gwadar Port are the Middle East and Near East, while those of Karachi port are (North- and South-) East Asia and those of Bin Qasim Port are Africa, Europe and North America. These regional differences are mainly caused by the differences in the GLS network with which each port is connected. In other words, the pattern of partner regions estimated in Gwadar Port is mainly based on the assumption that all the GLSs to call at Gwadar Port are regional services to connect with Middle East and South Asia. It should be noted that Pakistani ports (i.e., Karachi and Bin Qasim) share a significant portion of the partner ports for maritime containers handled in Gwadar Port in scenario *s6*. While some containers are considered to be transshipped in these ports, other containers are coming from their hinterland, such as CA countries, and are transported to Gwadar Port by domestic shipping. This is partly because the model assumes the use of MS at least once.

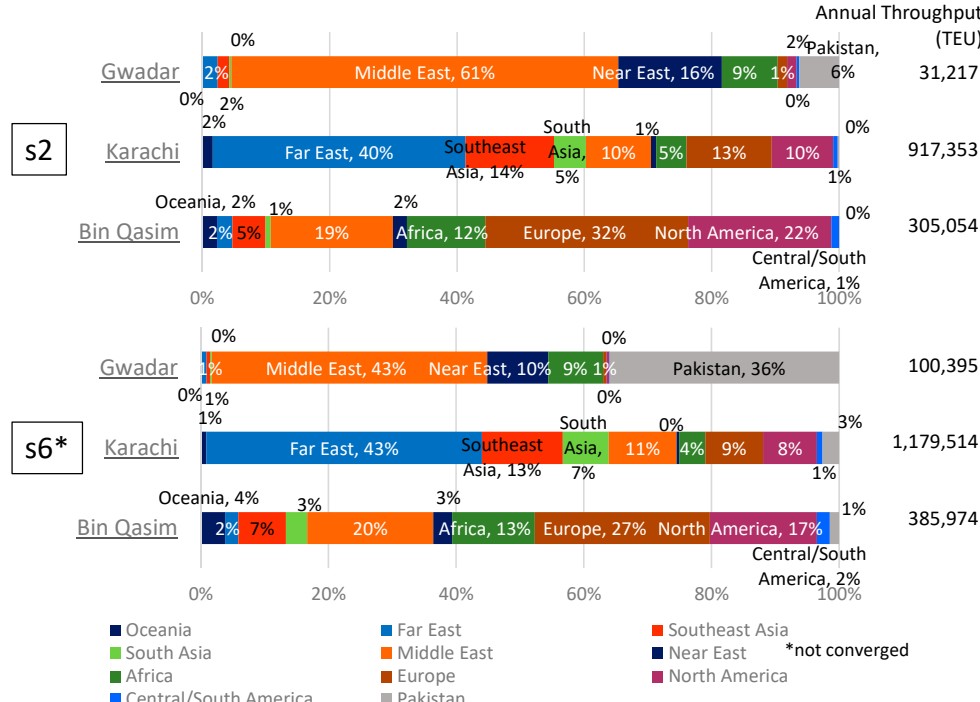

**Figure 13.** Regional breakdown of the partner ports for maritime containers in Pakistani ports in scenario *s2* and *s6*. Source: Compiled by the authors.

In summary, the above six scenario analyses show that Gwadar Port could handle over 30,000 TEUs annually, if the rail connectivity from the port was well developed, although some cargo would shift within the Pakistani ports. If the railway is connected not only with CA, but also with China, as in scenario *s4*, more containers (approximately 50,000 TEUs per year for laden containers) would use

Gwadar Port. These additional containers would shift partly from the gateway seaports in regions other than Pakistan. This implies that the improvement of the intermodality between ports and railways in Pakistan would only shift containers within the country, whereas Pakistan could offer gateway seaports to CA, if its international rail connections were also improved, especially those with China. Additionally, the impact on the throughputs would increase synergistically, if the border barriers decreased through an international cooperation (i.e., the CAREC programme) framework.

## 6. Conclusions

This study analysed the market potential of Gwadar Port and other Pakistani ports as gateways of the cargo to/from CA countries including China (the XUAR) and Russia (SFD). The results estimated with the model simulations revealed that Gwadar Port could handle a sustainable volume of containers, if the hinterland rail was well connected. If an international rail connectivity with CA via Afghanistan and China was available, Pakistani ports could play a key role as gateways for CA cargo, and the role would be intensified by lowering national border barriers under the CAREC framework. Particularly, improving the connections with China would obviously increase the cargo volume handled by Pakistani ports. In reality, the cooperation between China and Pakistan under the CPEC has been accelerated, including not only infrastructure investment in Pakistan and border facilitation between the two countries, but also the enhanced functionality of a Chinese city (Kashi), as a logistics hub. As the political instability in Afghanistan is unlikely to improve immediately, a Chinese route to connect CA with the Pakistani seaports is more realistic. Indeed, according to our interviews, some logistics companies and forwarders in CA are actively considering this option.

The major advantage of Pakistani ports is that they provide shorter distances, in terms of HT from most cities in CA, than other competing ports. If the infrastructure in and around the country was developed, and the barriers at the cross-borders were significantly reduced, the market potential of Pakistani ports, which enables the shifting of cargo from transport in the east–west directions to the north–south directions, would be realised. In addition, the three Pakistani ports, i.e., Karachi, Bin Qasim and Gwadar, have different partner regions. Particularly, Gwadar Port has a geographical advantage over the Middle East because of its MS network, although this partly depends on the GLS network with which each port is connected.

While this study showed empirical evidence regarding the market potential of Gwadar Port in terms of the container volumes drawn from the NAM, there are still many further issues to be resolved. First, a more detailed zoning system, especially for the larger countries (e.g., China and Russia) and for some European countries, is necessary to improve the model accuracy, using customs statistics and other sources in the region by country.

The second issue is the need for integrated modelling to cover both international MS and land transport cargo in the region. Herein, we considered only international maritime containers, which means that all cargo considered in the model are assumed to be moved by containerships at least once. This led to difficulties in handling the drastic change of cargo transport in this region from MS to wholly land transport by trucks or trains, and vice versa. To resolve this, the current land transport cargo demand, which is often not transported by containers, needs to be integrated in future studies.

A further issue is incorporating the networking of GLS into the NAM model. As the existing model does not consider the networking behaviour of international liner shipping companies (i.e., all GLSs are given), we should also preliminarily assume a change in the GLS network as a given input, as shown in Section 5, for the scenario involving Gwadar Port. It is difficult to check ex post facto whether such assumptions concerning the network change are reasonable. Internalising the behaviour of the liner shipping companies is another challenging issue.

**Author Contributions:** Conceptualization, R.S.; Data curation, R.S. and S.T.; Formal analysis, R.S. and S.T.; Funding acquisition, R.S. and H.K.; Investigation, R.S., S.T. and H.K.; Methodology, R.S. and S.T.; Project administration, R.S. and H.K.; Supervision, H.K. and P.T.-W.L.; Validation, R.S.; Visualization, R.S. and S.T.; Writing—original draft, R.S. and S.T.; Writing—review and editing, H.K. and P.T.-W.L.

**Funding:** This research was funded by Japan Society for the Promotion of Science (JSPS KAKENHI), grant number 17H03327 and 25289159.

**Conflicts of Interest:** The authors declare no conflict of interest.

## Appendix A. Summary of the Interview Survey and Field Trip

**Table A1.** List of interview surveys.

| Country | Place | Date | Interviewee | Interviewee's Organization |
|---|---|---|---|---|
| Pakistan | Karachi | July, 2015 | Ministry of Ports and Shipping | Government |
| | | | Karachi Port Trust (KPT) | Port administrator |
| | | | Karachi International Container Terminal (KICT) | Terminal operator |
| | | | Pakistan International Container Terminal (PICT) | Terminal operator |
| | | | DP World Karachi | Terminal operator |
| | | | Sojitz Corporation, Sumitomo Corporation | Cargo owners and Investors |
| | | | P.S.S Container, PAKLINK Shipping Services | Forwarders to Afghanistan |
| | Bin Qasim | | Qasim Port Authority (QPA) | Port administrator |
| | | | National Industrial Park (NIP) | Special Economic Zone (SEZ) |
| Kazakhstan | Astana | August, 2017 | Deutsche Gesellschaft für Internationale Zusammenarbeit (GIZ) | International organization (Germany) |
| | | | Ministry of Investment and Development | Government |
| | | | Kazakhstan Temir Zholy (KTZ) Express | Railway company (Container train operator) |
| | | | Marubeni Corporation | Cargo owner and Investor |
| | | August, 2018 | Kazakhstan Railway | Railway company |
| | | | Continental Logistics | Distribution Centre/ICD |
| | | | Embassy of Japan | Diplomatic organization |
| | Almaty | August, 2017 | Association of National Forwarders | Forwarder (railway) |
| | | | Globalink, Senko, Azuma Shipping, Nisshin | Logistics companies |
| | | | Mitsubishi Corporation | Cargo owner and Investor |
| | | August, 2018 | European Bank for Reconstruction and Development (EBRD) Almaty Branch Office | International organization |
| | | | Unico | Logistics companies |
| | | | Damu Logistics | Distribution Centre |
| | Khorgos | August, 2017 | International Center for Boundary Cooperation (ICBC) | Free Trade Zone (FTZ) |
| | | | Khorgos SEZ | SEZ, Dry port |
| | Aktau | August, 2018 | Aktau Port | Terminal operator |
| | | | Aktau North Port | Terminal operator |
| Uzbekistan | Tashkent | September, 2009 | State Customs Committee | Government |
| | | | Uzbek Agency of Road and River Transportation | Government |
| | | | State-Joint-Stock Railway Company (SJSRC) | Railway company |
| | | | Association of International Road Carriers | Logistics companies |
| | | | Association of International Forwarders of Uzbekistan | Forwarders |
| | | | Inland Container Depot | Dry Port (railway) |
| | | November, 2013 | Tashkent office, Japan External Trade Organization (JETRO) | Public organization |
| | | | Itochu Corporation | Cargo owner and Investor |
| Turkmenistan | Ashkhabad | November, 2013 | Embassy of Japan | Diplomatic organization |
| | | | Sojitz Corporation | Cargo owner and Investor |
| Kyrgyz | Bishkek | November, 2013 | Freight Operators Association of the Kyrgyzstan (KGZ FOA) | Logistics companies |
| Tajikistan | Dushanbe | August, 2017 | JICA Tajikistan office | International organization |
| | | | Ministry of Transport | Government |

**Table A1.** *Cont.*

| Country | Place | Date | Interviewee | Interviewee's Organization |
|---|---|---|---|---|
| Azerbaijan | Baku | November, 2013 | Transport Corridor Europe/Caucasus/Asia (TRACECA) | International organization (EU) |
| | | | Azerbaijan International Road Carriers Association (ABADA) | Logistics companies |
| | | | Itochu Corporation | Cargo owner and Investor |
| Georgia | Tbilisi | August, 2018 | JICA Georgia office | International organization |
| | | | Anaklia Development Consortium | Terminal developer |
| | | | Association of Freight Forwarders of Georgia | Forwarders |
| | Poti | | APM Terminals Poti | Terminal operator |
| China | Liangyungang | December, 2006 | Lianyungang Port Company | Port administrator/Terminal operator |
| | Urumqi | November, 2013 | Xinjiang Logistics Association | Logistics association/University |
| | | | Xinjiang Huacheng International Transportation service CO., LTD. | Logistics company |
| | | | DB Schenker Urumuch branch | Logistics company |
| Russia | Vladivostok | December, 2012 | Primorsky Krai Government | Local government |
| | | | Pacific Strategy Development Center, Regional Fund, Administration of Primorsky Territory | Governmental research institute |
| | | | Commercial Port of Vladivostok | Terminal operator |
| | | | Far Eastern Marine Research, Design and Technology Institute (FEMRI) | Research institute |
| | Vostochny | | Vostochnaya Stevedoring Company (VSC) | Terminal operator |
| | | | Vostochny Port | Terminal operator |
| | | | Rosmorport Vostochny Branch | Port administrator (landlord) |
| Japan | Tokyo | November, 2013 | ITS Nippon | Logistics company, Consultant |
| | | June, 2017 | Toyota Tsusho Co | Cargo owner and Investor |
| | | July, 2017 | Senko Co. | Logistics company |

Source: Compiled by the authors.

**Table A2.** Field visit list of land national borders in the CA region.

| Visiting Point | Date | Visiting Route | Used Transport |
|---|---|---|---|
| Ala Shankou (China)—Dostik (Kazakhstan) | 17 August 2008 | crossed from China to Kazakh | rail |
| Chorgos (China)—Khorgos (Kazakhstan) | 5 August 2017 | crossed from Kazakh to China | car |
| Irkeshtam (China)—Irkeshtam (Kyrgyz) | 7 August 2017 | crossed from China to Kyrgyz | car |
| Korday (Kazakhstan)—Akjol (Kyrgyz) | 18 August 2008 | crossed from Kazakh to Kyrgyz | car |
| | 6 November 2013 | accessed from the Kyrgyz side | car |
| Karasu (Kazakhstan)—Aktilek (Kyrgyz) | 6 November 2013 | accessed from the Kyrgyz side | car |
| Konsybaeva (Kazakhstan)—Yallama (Uzbekistan) | 2 October 2009 | accessed from the Uzbek side | car |
| | 6 November 2013 | accessed from the Uzbek side | car |
| Dustlik (Uzbekistan)—Osh (Kyrgyz) | 8 August 2017 | crossed from Kyrgyz to Uzbek | car |
| Jartepa (Uzbekistan)—Sarazm (Tajikistan) | 2 October 2009 | accessed from the Uzbek side | car |
| Andarkhon (Uzbekistan)—Patar (Tajikistan) | 8 August 2017 | crossed from Uzbek to Tajik | car |
| Alat (Uzbekistan)—Farab (Turkmenistan) | 30 September 2009 | accessed from the Uzbek side | car |
| | 5 November 2013 | crossed from Turkmen to Uzbek | car |
| Artyk (Turkmenistan)—Lotfabad (Iran) | 4 November 2013 | accessed from the Turkmen side | car |

Source: Compiled by the authors.

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
