# Peer review of "Could Gwadar Port in Pakistan Be a New Gateway? A Network Simulation Approach in the Context of the Belt and Road Initiative"

_sustainability, doi:10.3390/su11205757_

Round 1

Reviewer 1 Report

I evaluate the scientific paper as actual. I find the article both interesting and well written. Scope, methodology and results are presented satisfactorily. Authors used the scientific methods to evaluate the Gwadar Port's possibility of becoming a new gateway for CA cargo.

I have some suggestions for improvement the paper. In the chapter 2 (Literature review) It would be better specify in more detail the subject and result of author's review – literature 8-12 and 24-27.  I would appreciate the little discussions regarding security of rail/road transport across Pakistan and at the country's borders.

Globally, the article is written at a very good level. The authors used scientific methods and available information to obtain relevant results.

Author Response

Thank you for your kind comments.

(1) We add more detailed summary of literature 8–12 (in Lines 100–110) and 24–27 (in Lines 125–132).

(2) Detailed discussion on the security in Pakistan and its borders should appear on the future discussion, because it may contain many topics to be carefully considered. We add small sentences in Lines 173–175.

Reviewer 2 Report

Transport corridors analysis did not take in account TRACECA, which is important CA and Central and Western European Countries. In Article models and explanations did not take in account World and regional economics crises influence on cargo flows analysed in Article Not clear cargo flows forecast, would be important checked and recalculate cargo flows, which are taken from references. In formula (7) coefficient 0,75 it is always correct. This coefficient very much depends for the ocean and feeder shipping lines.    

Author Response

Thank you for your kind comments.

(1) According to your comment, we briefly mention about TRACECA in Lines 164–165.

(2) The model that the authors used in this paper focuses on the route and transport mode choice on the intermodal container shipping network. In other words, the cargo shipping demand is given as a model input. Therefore, another model is necessary for forecasting the impact on trade and cargo shipping demand patterns of the world and regional economic change. We add more explanation on this point in Lines 223–228.

(3) We add on the verification of the model agreement with the actual data in Lines 402–409 including one figure.

(4) It is difficult to estimate the exact coefficient on Eq (7) for each port due to lack of data. This point is additionally described in Lines 277–278.